

# Climatology of Lyapunov exponents: The influence of atmospheric rivers on large-scale mixing variability

Daniel Garaboa-Paz[1], Jorge Eiras-Barca[1], and Vicente Pérez-Munũzuri[1]

[1]Group of Nonlinear Physics. Faculty of Physics. University of Santiago de Compostela. 15782 Santiago de Compostela, Spain.

*Correspondence to:* D. Garaboa-Paz (angeldaniel.garaboa@usc.es) and V. Pérez-Munũzuri (vicente.perez@cesga.es)

**Abstract.**

Large-scale tropospheric mixing and Lagrangian transport properties have been analyzed for a long-term period 1979-2014 in terms of the finite-time Lyapunov exponents (FTLE). Wind fields reanalysis from the European Centre for Medium-Range Weather Forecasts were used to calculate Lagrangian trajectories of large ensembles of particles. The FTLE climatology shows large correlation values with the baroclinic instability growth rate. Larger values of the inter and intra-annual mixing variabilities highlight El Niño Southern Oscillation, the storm track or the Intertropical Convergence Zone among other large-scale structures. As a case study, the role that atmospheric rivers have on the large-scale atmospheric mixing and the precipitation rates observed in the Sahara-Morocco and British Isles regions have been analyzed. Atmospheric rivers contribution to tropospheric mixing is found to decrease from $15\%$ in Sahara-Morocco to less than $5\%$ for UK-Ireland regions, in agreement to their contribution to precipitation that is $40\%$ larger in the former than for the latter region.

## 1 Introduction

Large-scale tropospheric mixing and transport barriers to air masses play an important role to characterize weather. Together with the Coriolis effect and the distribution of the continents, the conversion between thermal into kinetic energy is the main triggering mechanism ruling the large scale atmospheric circulation. Extratropical cyclones and jets outside of the tropics, monsoons, and hurricanes in the tropics, among others, are the main structures for atmospheric mixing.

One approach to characterize mixing and transport is by calculating Lagrangian trajectories of passive tracers in the atmosphere. Among the different statistics that can be calculated (dispersion, diffusivity, etc), finite-time Lyapunov exponents (FTLE) measure the separation of two trajectories over time from initially nearby starting points, i.e. the local mixing rates at a finite time.

The link between transport and climate, in terms of long term statistics of Lagrangian quantities (*James*, 2003; *Stohl*, 2006), and the global climate change variability of atmospheric mixing (*Holzer and Boer*, 2001) has been studied. FTLE have been used to identify the presence of barriers to mixing in the atmosphere between the tropics and extratropics (*Pierrehumbert and Yang*, 1993), to study the zonal stratospheric jet (*Beron-Vera et al.*, 2008), jet-streams (*Tang et al.*, 2010), hurricanes (*Rutherford et al.*, 2012), transient baroclinic eddies (*von Hardenberg and Lunkeit*, 2002), the spread of plankton blooms (*Huhn et al.*, 2012)



or the polar vortex (*Koh and Legras*, 2002). The predictability of the atmosphere for long periods of time has also been studied using FTLE (*Yoden and Nomura*, 1993; *Huber et al.*, 2001; *Stohl*, 2001; *Garny et al.*, 2007; *d'Ovidio et al.*, 2009; *Ding et al.*, 2015). Moreover, identification of ridges of maximum FTLE (*Shadden et al.*, 2005) allows the detection of potential Lagrangian coherent structures or kinematic transport barriers that control the flow mixing and folding over a period of time

for the examples cited above.

Considering all sources of large-scale atmospheric mixing necessary for a detailed mixing climatology would be overwhelming. In this Letter, we will concentrate on the effects of baroclinic instability and the accompanying effect of the advective moisture transport from the tropics and subtropics in tropospheric mixing. One of the main mechanisms that addresses the transport of air masses within the troposphere in mid-latitudes is baroclinicity (*Lindzen and Farrell*, 1980; *Hoskins and Valdes*,

1990). These regions are dominated by cyclone and anticyclone activity increasing tropospheric mixing, in contrast to tropical and subtropical latitudes. Atmospheric or tropospheric rivers (ARs) have been shown to play a key role in extratropical trophospheric dynamics (*Newel et al.*, 1994; *Zhu and Newell*, 1998; *Gimeno et al.*, 2016). These structures are narrow and elongated filaments that transport moisture from the tropics into mid-latitudes over a period of a few days, once a baroclinic structure develops. For some ARs events, a filament pattern develops living enough time to be considered as a Lagrangian coherent

structure (*Garaboa-Paz et al.*, 2015). The advection and convergence of moisture by ARs is a key process for the Earth's sensible and latent heat redistribution and has a strong impact on the water cycle of the mid-latitudes increasing tropospheric mixing. Additionally, the importance of a better understanding of ARs is beyond all doubt, since they have been shown to be closely related to extreme precipitation and flooding events in different parts of the world (*Dettinger et al.*, 2011; *Ralph et al.*, 2011; *Lavers et al.*, 2013; *Eiras-Barca et al.*, 2016).

The aim of this work is to investigate the long-term variability in tropospheric mixing using the FTLE, focusing on the role that large-scale structures with a timescale of days play on the global horizontal transport in the lower troposphere. To that end, we have calculated a climatology of FTLE for the period $1979--2014$ using wind fields retrieved from the European Center for Medium-Range Weather Forecast (ECMWF) reanalysis, ERA-Interim (*Dee et al.*, 2011). Intra and inter-annual changes in the FTLE time series over this long term period of time have been studied. We show that the mean FTLE and its variability

reveal inhomogeneities in mixing determined by regions of strong or weak mixing and barriers to air exchange. In addition, the contribution of land falling ARs to atmospheric mixing was found to decrease from a $15\%$ in Sahara-Morocco to a $5\%$ for the British Isles, in agreement with a larger contribution to precipitation in the southern region.

## 2   Data and Methods

The atmospheric transport has been studied using wind field data retrieved from the European Center for Medium-Range

Weather Forecast reanalysis, ERA-Interim (*Dee et al.*, 2011), with a horizontal spatial resolution of $0.7°$, a vertical resolution of 100 hPa and a temporal resolution of 6 hours.



In a longitude-latitude-pressure coordinate system $(\phi, \theta, P)$, the position of an air particle is calculated as,

$$
\begin{aligned}
\dot{\phi} &= \frac{u}{R\cos(\theta)} \\
\dot{\theta} &= \frac{v}{R} \\
\dot{P} &= w(\phi, \theta, P, t)
\end{aligned}
\tag{1}
$$

where, $u$, $v$ and $w$ are the eastward, northward and vertical wind components, respectively, and $R \approx 6370$ km is the Earth's mean radius. A fine grid of particles with an initial separation of $0.35°$ is uniformly distributed on the 850 hPa level covering the domain $(\theta, \phi) \in [0, 360] \times [-85, 85]$ at time instant $t_0$. Then, 3D Lagrangian simulations have been performed so that particle trajectories are computed integrating Eq. (2) using a 4-th order Runge-Kutta scheme with a fixed time step of $\Delta t = 1.5$ hours, and multilinear interpolation in time and space.

In order to characterize the atmospheric transport, we introduce the finite-time Lyapunov exponents (FTLE), that measure, at a given location, the maximum stretching rate of an infinitesimal fluid parcel over the interval $[t_0, t_0 + \tau]$ starting at $\mathbf{r}(t_0) = \mathbf{r_0}$ and ending at $\mathbf{r}(t_0 + \tau)$ (*Shadden et al.*, 2005; *Sadlo and Peikert*, 2007). The integration time $\tau$ must be predefined and it has to be long enough to allow trajectories to explore the coherent structures present in the flow. The FTLE fields $\lambda$ are computed along the trajectories of Lagrangian tracers in the flow as (*Peacock and Dabiri*, 2010),

$$
\lambda(\mathbf{r}_0, t_0, \tau) = \frac{1}{|\tau|} \log \sqrt{\mu_{max}(\tilde{\mathbf{C}}(\mathbf{r}_0))},
\tag{2}
$$

where $\mu_{max}$ is the maximum eigenvalue of the pull-back Cauchy-Green deformation tensor $\tilde{\mathbf{C}}$ over a sphere (*Haller and Beron-Vera*, 2012) which does not take into account the deformation due to vertical movement. Repelling (attracting) coherent structures for $\tau > 0$ ($\tau < 0$) can be thought of as finite-time generalizations of the stable (unstable) manifolds of the system. These structures govern the stretching and folding mechanism that control flow mixing.

Ridges in the FTLE field are used to estimate finite time invariant manifolds in the flow that separate dynamically different regions, and organize air masses transport. A positive time direction (forward FTLE) integration leads to identify lines of maximal divergence of air masses. In contrast, a negative time direction integration, leads to identify areas of maximal convergence (backward FTLE).

The time series of the FTLE field has been computed by following the same steps explained previously, but varying the initial time $t_0$ in fixed steps $\Delta t_0 = 6$ hours in order to release a new initial tracer grid. Each FTLE field obtained for each advection from $[t_0, t_0 + \tau]$ is an element of the time series $\lambda_i = \lambda(\mathbf{r_0}, t_0 + i\Delta t_0, \tau)$. FTLE are computed forward ($\tau > 0$) and backward ($\tau < 0$) time direction, so two time series have been generated. The finite integration times were chosen within the range $\tau \in [1, 15]$ days.

## 3   Results

We have studied the transport of air masses in terms of their FTLE from a climatological point of view. Figure 1(a) shows the backward FTLE for a given time at 850 hPa over the ocean. The structures reflect the large scale advection of air masses,





which are stretched and folded as wind transports them. The presence of ridges correspond to attracting manifolds where fluid tends to converge. Time averaged FTLE maps for the $1979 - 2014$ period are shown in Figs. 1(b,c) for forward and backward integration times, respectively. As it was expected, in both cases, three latitudinal bands can be clearly identified in coincidence with the large scale atmospheric circulation belts. For mid-latitudes, FTLE values are approximately twice higher than for the

equatorial zone. A clear annual cycle is observed, and in the mid-latitudes mixing is generally higher in winter than in summer (Figures S1 and S2 in the supporting information). Note as well that there is some longitudinal variability in the mixing that arises from the longitudinal variability in the Lyapunov exponents.

Focusing on the high-middle latitudes of the forward in time mean FTLE maps, the signal of global pressure systems can be identified. In the northern hemisphere we can observe two plumes with high FTLE values over the Atlantic and Pacific Ocean

that correspond to the storm track, leading to an increase of mixing and dispersion. The same situation arises in the southern hemisphere. Moreover, the large-scale subtropical centers are apparent as elongated tongues of low values of FTLE extending from the equatorial zone to the west of continents. These regions contain low FTLE values and correspond to low mixing regions.

The mean backward FTLE field shows smaller values in Fig. 1(c) for high latitudes than in the forward case. Thus, the

width of the region with low FTLE values near the Equator is larger than for the forward case. Low FTLE backward regions correspond to zones where the convergence of air masses to the Equator weakens.

Baroclinic instability is the dominant mechanism triggering the dynamics of mid-latitude weather systems. The largest values of the mean FTLEs have been obtained for mid latitudes in both hemispheres indicating an increase of atmospheric mixing in those regions. To further quantify the connection between mixing and baroclinicity, the Eady growth rate (*Lindzen and Farrell*,

1980; *Hoskins and Valdes*, 1990) has been calculated for the 850 hPa level as,

$$\sigma_{BI} = 0.31 \frac{|f|}{N} \left| \frac{\partial \mathbf{V}}{\partial z} \right| \tag{3}$$

where $f$ is the Coriolis parameter, $N$ is the Brunt-Väisälä frequency, $\mathbf{V}$ is the 3D wind component and $z$ is the geopotential height.

Figure 2(a) shows the time average Eady growth rate as a gridded map for the 850 hPa level for the $1979 - 2014$ period in

units of day$^{-1}$. Note the storm track regions (such as the North Atlantic or Norh Pacific corridors) are well depicted by this measure of baroclinicity, and if compared with the mean forward in time FTLE map, Fig. 1(b), both figures are remarkably similar. In order to quantify this coincidence, the correlation between both fields have been calculated for different $\tau$, Fig. 2(b). A correlation maximum is observed for an integration time of 5 days, which is about the mean length of the typical synoptic time scale, in line with the mean lifetime of extratropical cyclones (e.g. *Trigo*, 2006). Thus, the large values of atmospheric

mixing observed at mid-latitudes can be related, at least in part, to baroclinic instability.

To gain insight into the transport of air masses, the variability of the FTLE climatology has been studied in terms of the intra-annual (standard deviation of the monthly means for the 35-years) and inter-annual (standard deviation of the annual means for the 35-years) variabilities, Fig. 3. Regions where the FTLE change between seasons correspond to a large intra-annual variability. On time scales shorter than seasonal, variability of the circulation is dominated by synoptic weather systems, which



**Table 1.** Percentage of AR days and its associated precipitation rates out of the total for two Atlantic regions.

|  | **Sahara-Morocco** | **British Isles** |
| --- | --- | --- |
| **AR days** | 10.3% (1201 days) | 32.5% (3800 days) |
| **Precipitation** | 16.8% | 37.5% |

prevail at mid-latitudes. The forward in time intra-annual variability, Fig. 3(a), highlights the meridional frontier between westerly extratropical circulation and Hadley cells where larger variability is observed between seasons. As an example, note in the Pacific Ocean the plume of high variability observed that connect the semi-permanent pressure system between the Aleutian Low and the North Pacific High. A similar situation can be observed between the Iceland Low and the Azores High

for the Atlantic Ocean. As well, note the signal of the monsoons in the Indian Ocean.

The intra-annual variability map obtained from the FTLE backward time series, Fig. 3(b), shows regions with maximum variability through the year in the tropics. The main global mechanism which address this variability is the meridional movement of the Intertropical Convergence Zone (ITCZ). Note the importance of this variability in the African coast or the western Pacific Ocean. The interface between summer and winter ITCZ coincides with a region close to the equator with small variability.

Figure 3(c,d) shows the inter-annual variability calculated forward and backward in time, respectively. The inter-annual variability takes into account the variation through the 35 years of FTLE computed. In this case, both, forward and backward fields behave in a similar way although some differences are observed. All periodic effects are canceled out, and the El Niño Southern Oscillation (ENSO) pattern in the Pacific Ocean is shown in the backward map. Although easterly trade winds converging across the equatorial Pacific weaken during El Niño phase, during La Niña and neutral conditions those winds are

reinforced, and the inter-annual backward FTLE values should be larger in the western Pacific (d). However, for the forward case (c), injection of Lagrangian particles into the equator zone propagates with the converging trade winds and few dispersion areas within the Tropics are observed. For the analyzed climate period, the variability introduced by this region is approximately 10% of the global mean FTLE.

Comparing the inter-annual and intra-annual scales, the values of intra-annual scale are clearly higher than inter-annual

variability in the extratropical zone, however this difference is reduced in the equator zone except for some zones of the western Pacific due to ENSO.

Another important source of mixing in the atmosphere are the atmospheric rivers (ARs) that play a key role in baroclinic dynamics. ARs appear in mid-latitudes as coherent filaments of water vapor triggering atmospheric mixing and the convergence of moisture in the lower levels of the troposphere with a persistence time of several days up to a week (*Garaboa-Paz et al.*,

2015). As a case study, we have focused on the contribution of ARs to atmospheric mixing along the $1979 - 2014$ period for two Atlantic regions: Sahara-Morocco and the British Isles.

Figure 4 shows the FTLE backward time average computed only when ARs are detected divided by the mean backward FTLE over Sahara-Morocco (a) and the British Isles (b), using daily-AR landfall detection criteria provided by *Guan and Waliser* (2015). This quantity leads to identify regions where the ARs activity has a major role over the climate background



in terms of backward FTLE. Since the FTLE ratio signal is clearly stronger for the African case, ARs should play a more prominent role in the large-scale mixing and convergence of moisture in the Sahara's coast than in the British one. Therefore, this idea should be consistently abided when precipitation is taken into account. Table 1 shows the rainfall during AR events out of the total at each of both regions (see as well Figure S3 in the supporting information). Even when AR detections are

more frequent in the British Isles (32.5% of the days) than in Sahara-Morocco (10.3% of the days), the contribution of ARs to precipitation in Sahara-Morocco is 41.7% larger than for the British Isles. The Sahara-Morocco region has lesser ARs activity than UK-Ireland but the contribution to precipitation is more important, in agreement with a larger anomaly in the FTLE backward mixing ratio.

## 4   Conclusions

The finite-time Lyapunov exponents (FTLE) time series at 850 hPa level has been computed over a climate period of 35 years using wind fields retrieved from ERA-Interim reanalysis data. The FTLE provide information on areas where the dispersion (integration forward in time) or the convergence (backward) is large and allow classification of airstreams. The statistics over these Lagrangian quantities have shown the link between the climate system and the regional transport structures in terms of atmospheric mixing.

This study, one of the first to present a 35 year of large-scale atmospheric mixing, shows mean values, and intra-annual and inter-annual variability of the FTLE. To show the mixing effects more clearly, forward and backward in time FTLE fields were obtained. Mean Lyapunov exponents show a zonal localization; large values in the mid-latitudes for both hemispheres, while the lowest FTLE values were observed inter tropics. Especially in the tropics and Equator, mixing is strongly modulated by ENSO, while for mid-latitudes, large-scale mixing is associated to the interface between westerly extratropical circulation and

Hadley cells. The meridional displacement of the ITCZ has also been well reproduced by the intra-annual backward FTLE field. Seasonal effects and ENSO are the largest effects that contribute to large-scale mixing variability over the globe.

The mean FTLE field was correlated to the Eady baroclinic growth rate. It was found that the best correlation is obtained for an integration time of $\tau = 5$ days, which is in agreement with the typical synoptic time scale in mid-latitudes. For larger time scales, structures observed in the intra-annual and inter-annual variability fields are smeared out, while for smaller $\tau$ values

those structures are not well shaped, and multiple patterns arise. This suggests that baroclinicity, among other possible causes, drives large-scale atmospheric mixing on time scales longer than a few days.

As an example of air masses transport, we focused on the impact of atmospheric rivers (ARs) on mixing in the Atlantic Ocean. The advection of moisture by ARs is a key process for the Earth's sensible and latent heat redistribution and has a strong impact on the water cycle of the mid-latitudes. In a previous work we found that these structures can be well described

in terms of the FTLE (*Garaboa-Paz et al.*, 2015). Here, we find that the impact of mixing in the Sahara-Morocco region is more important than for the British Isles. Although less ARs and low precipitation rates are observed in the Sahara-Morocco if compared to UK-Ireland, rain probability during ARs events and mixing are larger for the former than for the latter region. Finally, our results suggest that atmospheric mixing, as shown in terms of large FTLE values, correlate well with baroclinic



instability and could be used to forecast precipitation events in those regions where the persistence of coherent transport structures has a great impact.

*Acknowledgements.* ERA-Interim data were supported by ECMWF. This work was financially supported by Ministerio de Economía y Competitividad and Xunta de Galicia (CGL2013-45932-R, GPC2015/014), and contributions by the COST Action MP1305 and CRETUS Strategic Partnership (AGRUP2015/02). Computational part of this work was done in the Supercomputing Center of Galicia, CESGA. We acknowledge fruitful discussions with S. Brands G. Míguez, and we would like to thank Dr. Bin Guan for kindly sharing the ARs database.





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







**Figure 1.** Backward finite-time Lyapunov exponents λ for a given day (a). Local maxima in the plot (darker colors) are attracting coherent structures. Mean forward (b) and backward (c) FTLE climatology for the 1989-2014 period. For all cases, $\tau = 5$ days.





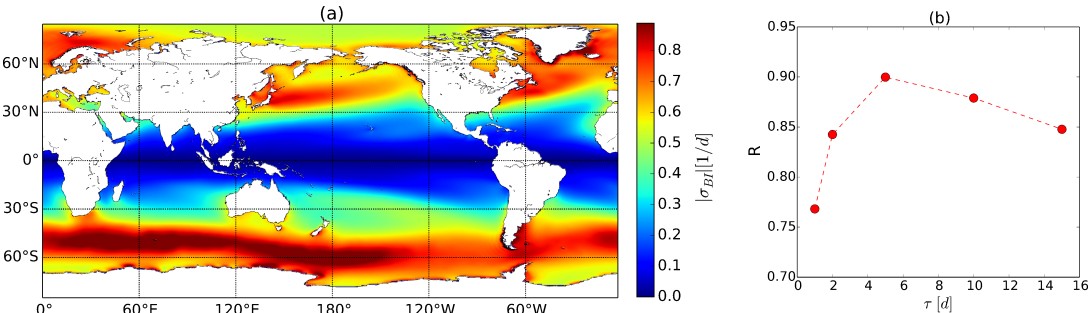

**Figure 2.** (a) Time average for 35 years of the baroclinic Eady growth rate, Eq. (3), calculated at 850 hPa. (b) Correlation index $R$ between $\sigma_{BI}$ and the 35-year time average forward FTLE map shown in Fig. 1(b) for different integration times $\tau$.

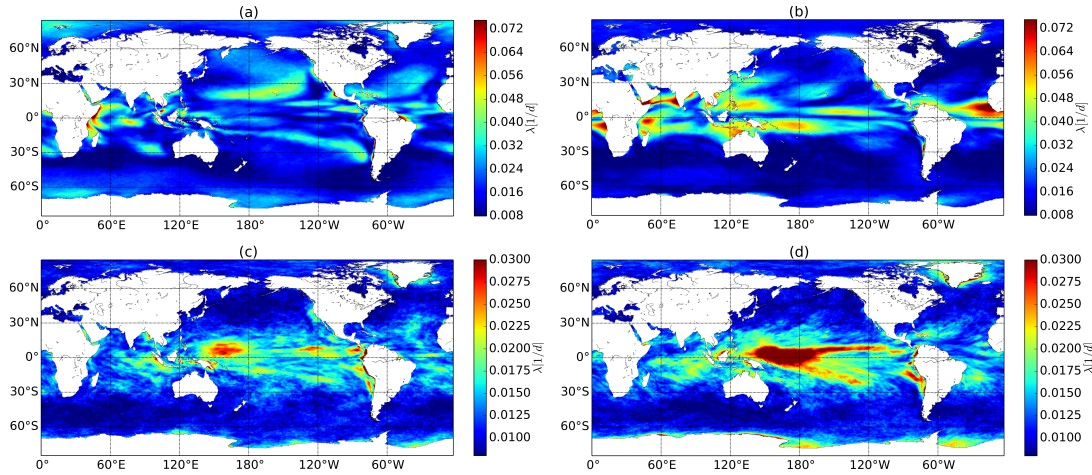

**Figure 3.** Seasonal dependence of the Finite-Time Lyapunov Exponents calculated for the 1979-2014 period. Intra-annual variability of the forward (a) and backward (b) FTLE, respectively. Inter-annual variability of the forward (c) and backward (d) FTLE, respectively. For all cases, $\tau = 5$ days.



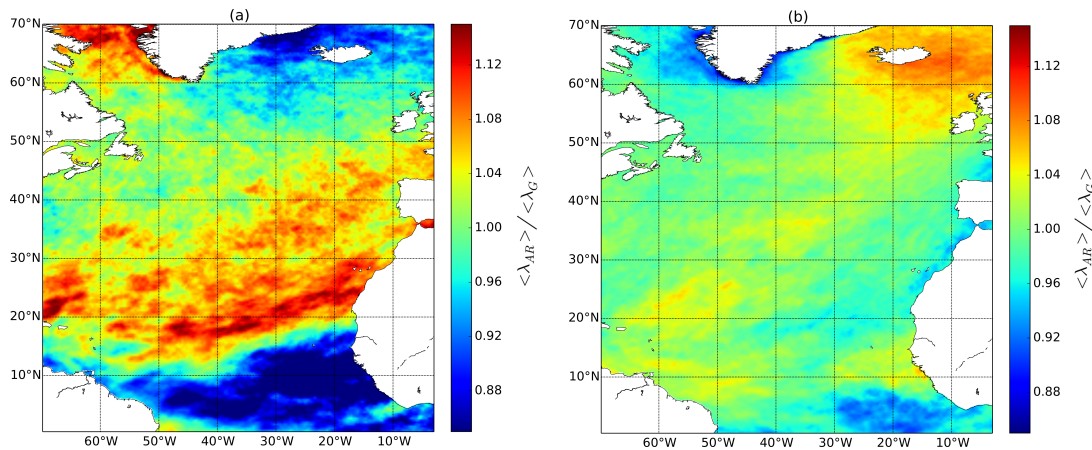

**Figure 4.** Ratio of the FTLE backward time series consisting of periods with land falling atmospheric rivers and the global backward FTLE mean (Fig. 1(c)) for the Sahara-Morocco (a) and UK-Ireland (b) regions.