# Peer review of "Climatology of Lyapunov exponents: The influence of atmospheric rivers on large-scale mixing variability"

_Earth System Dynamics, 2017_

## Referee Comment (RC1) · Anonymous Referee #1 · 27 Feb 2017

This paper studies mixing and Lagrangian transport properties for a period of 35 years, 1979-2014, of the wind field reanalysis from the ECMRWF by computing trajectories of a large number of tracers placed in a grid of 0.35 degrees. Lagrangian simulations are carried out using the wind data as input and FTLEs are calculated for each tracer on a time horizon  $\tau$ . Potential connections of FTLEs maps with baroclinic instability, ENSO, storm tracks, etc are discussed. Some of these connections/correlations are clearer than others, some are put in firmer grounds than others, and some are no more than a conjecture, but the paper has potential to be interesting.

The paper has however some scientific issues that should be addressed:

1) The FTLE definition in Eq.(2) cannot be correct. The deformation tensor C must

depend on  $t_0$  and  $\tau$  explicitly. I guess the authors mean C is the product of deformation tensors evaluated along the trajectory of the tracer from  $t_0$  up to  $t_0 + \tau$  at every time step in the integration. Ideally, one should write the explicit equations that go from the motion equations to  $\lambda$  to make the paper accessible to a wider audience– namely, those who are not specialists in Lagrangian flows. In any case, the formula (2) must be corrected and the correct meaning for C must be given.

2) Below line 25, the paragraph that begins: "Figure 4 ..." discusses ARs (atmospheric Rivers) the authors mention they use some detection criteria by Guan and Waliser and nothing else is explained. Well, I don't think this method is that well known to a general audience so that everyone should know how ARs were actually detected. One does not know why this method is used and no others or how would that change detection. The explicit details of how this detection works, why is favoured by the authors here, etc should be provided.

3) I do not know how periods with land falling ARs are calculated and I fail to fully appreciate the validity of Fig. 4. What does it mean  $\lambda_{AR}$ ? You mean the FTLE is only computed during those episodes of AR events? Does this mean the whole interval  $(t_0, t_0 + \tau)$  must be within the event? Or only the starting time  $t_0$ ?

4) The last sentence of the paper is intriguing. When the authors say: "... and could be used forecast precipitation events in those regions where persistence of coherent transport structures has a great impact", do they really mean to say FTLEs can be used to *forecast precipitation events*?

Typos:

1) In the first sentence of the paper I think "the conversion of" is better than "the conversion between"

2) In Page 3, line 8: It should be Eq.(1) instead of (2)

3) Page 3, line 18: "stable (unstable)" shouldn't it be "unstable (stable)"?

СЗ

---

## Author Comment (AC1) · 20 Apr 2017

We would like to thank the Referee for his/her valuable comments and critics that we tried to take into account in the revised version of the manuscript. Hopefully, all the major and minor corrections pointed out by the reviewer have been corrected now. A detailed answer follows below. We provide replies to the reviewer' comments in bold. As well, corrections included in the manuscript are marked in red.

Answer to Referee 1

This paper studies mixing and Lagrangian transport properties for a period of 35 years, 1979-2014, of the wind field reanalysis from the ECMRWF by computing trajectories of

a large number of tracers placed in a grid of 0.35 degrees. Lagrangian simulations are carried out using the wind data as input and FTLEs are calculated for each tracer on a time horizon  $\tau$ . Potential connections of FTLEs maps with baroclinic instability, ENSO, storm tracks, etc are discussed. Some of these connections/correlations are clearer than others, some are put in firmer grounds than others, and some are no more than a conjecture, but the paper has potential to be interesting.

The paper has however some scientific issues that should be addressed: 1) The FTLE definition in Eq.(2) cannot be correct. The deformation tensor C must C1 depend on t0 and  $\tau$  explicitly. I guess the authors mean C is the product of deformation tensors evaluated along the trajectory of the tracer from t0 up to t0 +  $\tau$  at every time step in the integration. Ideally, one should write the explicit equations that go from the motion equations to  $\lambda$  to make the paper accessible to a wider audience– namely, those who are not specialists in Lagrangian flows. In any case, the formula (2) must be corrected and the correct meaning for C must be given.

We agree with the referee on this insight. We rewrite the methods section to make clear the FTLE calculation; also we included the explicit dependences. The Cauchy tensor is not evaluated each time step. It is just evaluated when particles reach their final position at  $t0+\tau$ .

2) Below line 25, the paragraph that begins: "Figure 4 ..." discusses ARs (atmospheric Rivers) the authors mention they use some detection criteria by Guan and Waliser and nothing else is explained. Well, I don't think this method is that well known to a general audience so that everyone should know how ARs were actually detected. One does not know why this method is used and no others or how would that change detection. The explicit details of how this detection works, why is favoured by the authors here, etc should be provided.

The AR-Detection Database provided by Guan and Waliser is the most widely used database nowadays. We didn't just follow the detection criteria, but we directly used the
database, which is public. We have included more information about the AR-detection method in the paper. Specifically: We have changed "using daily-AR landfall detection criteria provided by Guan and Waliser (2015)" by:

The AR landfall detection has been carried out using the AR-Database provided by Guan and Waliser (2015). This database identifies ARs by complex considerations on the continuity and coherence of the integrated water vapor column and water vapor flux. Since it is able to identify ARs throughout the year and worldwide, this database provides, to the best of our knowledge, the most complete AR database published nowadays [Waliser and Guan (2017)].

3) I do not know how periods with land falling ARs are calculated and I fail to fully appreciate the validity of Fig. 4. What does it mean  $\lambda$ \_AR? You mean the FTLE is only computed during those episodes of AR events? Does this mean the whole interval (t0, t0 +  $\tau$ ) must be within the event? Or only the starting time t0?

The AR periods are calculated based on the true detection method coming from the database (Guan, 2015) mentioned previously. The procedure is the following: 1. Using the Guan database, we build a true-false time series based on the presence of ARs over the region of interest. 2. We use this mask to select the FTLE maps time steps with a true AR detection. 3. We apply the mean over the true elements (2) obtaining  $\lambda$ \_AR. To avoid misunderstandings we have added new sentences in the text to clarify this point.

4) The last sentence of the paper is intriguing. When the authors say: "... and could be used forecast precipitation events in those regions where persistence of coherent transport structures has a great impact", do they really mean to say FTLEs can be used to forecast precipitation events?

This method cannot replace the weather forecast simulations. As we comment in the methods section, the FTLE can be obtained in forward and backward direction. To compute the FTLE in backward direction we just need information from the wind field,

**ESDD**
from the past to the present. Performing, backward advections, we can estimate the presence of attracting coherent structures in the wind field. If an attracting coherent structure starts to develop and there are precedents of a similar dynamical behaviors (like ARs), this information can be used to estimate how the deformation of air masses will be transported in the following days.

Typos: 1) In the first sentence of the paper I think "the conversion of" is better than "the conversion between" 2) In Page 3, line 8: It should be Eq.(1) instead of (2) 3) Page 3, line 18: "stable (unstable)" shouldn't it be "unstable (stable)"?

Thank you to indicate us these typos that we have corrected. With respect to the last comment, repelling coherent structures can be thought as stable manifolds and vice versa.

Please also note the supplement to this comment: http://www.earth-syst-dynam-discuss.net/esd-2017-1/esd-2017-1-AC1supplement.pdf

---

## Author Comment (AC2) · 20 Apr 2017

We would like to thank the Referee for his/her valuable comments and critics that we tried to take into account in the revised version of the manuscript. Hopefully, all the major and minor corrections pointed out by the reviewer have been corrected now. A detailed answer follows below. We provide replies to the reviewer' comments in bold. As well, corrections included in the manuscript are marked in red.

Answer to Referee 2

The paper provides an analysis of low tropospheric mixing (850 hPa) in terms of finite-time Lyapunov exponents computed from the European Centre for Medium-Range

[Figure]

Weather Forecasts (ECMWF) Era-Interim dataset for the period 1974-2014. Two main results are provided. The first one links Lyapunov exponents to the baroclinic growth rate. The second result is a link between Lyapunov exponents and atmospheric rivers. The paper seems to have some potential, but I have difficulties in assessing its quality, because of the reasons discussed below. First of all, I find the paper very short, in particular for the Results section. The result about the impact of Atmospheric Rivers (ARs) and mixing, which gives the title to the paper, takes 15 lines in the Results Section, and is then discussed in even less lines in the Discussion.

We agree with the reviewer that the text dedicated to ARs was too small. We have increased the Results section adding a new Figure that allows us to identify the role played by ARs in large-scale atmospheric mixing. The new figure 4 is strikingly similar to Fig.1b and Fig.2a as the three of them account for the main sources of mixing in mid-latitudes; baroclinicity and ARs, and both are characterized in terms of the FTLEs. We have also modified the paper's title as it was confused, invoking the idea that ARs were the main subject of the paper.

The other result, on the link between baroclinic instability and Lyapunov exponents, takes only a bit more space. For a reader like me, who is not a specialist in atmospheric processes but interested in more general subjects like geophysical mixing, it is very difficult to appreciate the importance of the results, as well as the motivation of some of the choices, like for instance the regions in the case studies. This objection I think is important for ESD, which promotes interdisciplinary research on the Earth system in general.

Baroclinic instability is the dominant mechanism triggering the dynamics of mid-latitude weather systems. It shapes the cyclones and anticyclones that dominate weather in mid-latitudes and cause most of the large-tropospheric mixing in those latitudes. As FTLE are a measure of mixing in a flow, it seems appropriate to relate both quantities in the context of this paper. A measure of baroclinic instability can be obtained in terms of the Eady growth rate. The obtained correlation between both quantities confirms that
(i) FTLE can describe large-scale mixing in the troposphere, and (ii) the best correlation is obtained for an integration time of $\tau =5$ days, which is in agreement with the typical synoptic time scale in mid-latitudes.

Why comparing specifically Sahara-Morocco and the British Isles? Are they representative of other larger systems? How this result can be interpreted, or used in other studies?

We do not intend to procure general conclusions about the comparison of different latitudes in this regard that would involve a complete analysis which is out the scope of this paper. Our intention is only to provide a case study showing the role of precipitation in this issue, and how conclusions regarding to rainfall are consistent with those obtained throughout the paper.

Does the link between baroclinic instability and Lyapunov exponent address a specific knowledge gap, or it is an incremental result, or a confirmation?

We could consider it an incremental result and a confirmation. The trigger mechanism which mixes the atmosphere at the synoptic scales is the baroclinicity. We couldn't find any studies where the link between mixing on the troposphere, baroclinicity and AR activity has been quantified.

What are the challenges in atmospheric science that can benefit from the results of this paper? Most of the climatology atmospheric research is done based on Eulerian quantities. In this paper, we have used Lagrangian quantities to quantify mixing and dispersion at synoptic timescales over the troposphere.

Using Lagrangian quantities with a fixed time scale, allow us to identify how a particular timescale structure in the troposphere is affected by the climate and, at the same time, we can quantify the role that these structures have on the climate. This cannot be reproduced with Eulerian quantities. This paper pretends to address this challenge.

The paper should be strengthened in all of its parts: in the Introduction, to motivate

more the approach; in the Results, to motivate more the specific choices; and in the Conclusions, to discuss the possible larger impact of the results in terms of the challanges presented in the Introduction. For instance, some information is given about the relevance of Atmospheric Rivers (lines 15-20). Probably because of my (lack of) background, to me however is difficult to understand how the result of this paper specifically contributes to our understanding of the open questions related to ARs. Does this work really advocate as the main perspective the use of Lyapunov exponents for forecasting precipitations in some regions?

This method cannot replace the weather forecast simulations. As we comment in the methods section, the FTLE can be obtained in forward and backward direction. To compute the FTLE in backward direction we just need information from the wind field, from the past to the present. Performing, backward advections, we can estimate the presence of attracting coherent structures in the wind field. If an attracting coherent structure starts to develop and there are precedents of a similar dynamical behaviors (like ARs), this information can be used to estimate how the deformation of air masses will be transported in the following days.

My second remark is methodological, and is about the choice of the pressure level (850 hPa). The manuscript mentions tropospheric mixing, but in fact only low tropospheric mixing is analysed. This fact rises some questions: - What are the reasons behind the choice of the 850 hPa value? - As far as I understand, atmospheric rivers are not located in the low troposphere only. What are the arguments by which mixing at higher pressure levels can be neglected? What is the effect on the conclusions? - What are the limitations for studying baroclinic growth rates at 850 hPa only?

We want to focus on the troposphere, but at the same time we wanted to avoid the atmospheric events close to the surface within the PBL. We are interested in the large-scale tropospheric mixing. To that end, we start the advection at the intermediate level of 850hPa so the observed coherent structures are not perturbed by turbulence effects coming from the PBL. We performed integrations at different levels 850 hPa, 500 hPa

and 300 hPa. Although the main synoptic coherent structures remain qualitatively the same (see for example 3D simulations done in (Garaboa-Paz et al, 2005) at different levels, FTLE ridges diminish as pressure decreases. Moreover, in some cases, as pressure decreases, we notice that some structures become weaker and the integration time should be modified to capture these structures with larger resolution. The baroclinic growth rates are calculated using data from 1000 hPa to 750hPa to compute the finite differences concerning the potential temperature and wind stresses. Figure 2(a) shows the obtained results and the areas with larger baroclinic instability (mostly in mid-latitudes) are well reproduced.

Summing up, I find the paper with some potential but I feel that the text should be extended, or if the format does not allow, at least consolidated. The presentation in particular should be improved, and aimed at establishing a more solid link between the motivations and the perspectives opened by the results. Regarding the analysis, the authors should also provide more arguments for the fact that their calculation is limited to 850 hPa, but the outcome used for discussing phenomena occurring in a region which a much larger vertical extension.

As far as we know, no climatological studies have been done to identify the main sources of large-scale tropospheric mixing in the atmosphere. Inter and intra-annual variabilities of the FTLEs were able to reproduce the main synoptic large-scale structures as El Niño Southern Oscillation, the storm track or the Intertropical Convergence Zone among others. As specific examples for mid-latitudes, the influence of baroclinic instability and atmospheric rivers on tropospheric mixing has also been studied. On the other hand, Abstract, Introduction and Conclusions have been reinforced to strengthen the connection between baroclinic instability, atmospheric rivers and the large-scale mixing measured in terms of the FTLE.

Other comments: Convection: Convection can play a strong role at 850 hPa. How is convection taken into account, or what are the reasons for which it is neglected by the advection scheme?

The particles are advected in 3D, but, only the deformation due to 2D horizontal movement is considered. We want to focus on the horizontal spatial deformation instead of vertical deformation. The vertical-horizontal scales are completely different in the atmosphere, so considering the deformation due to vertical movement will lead to define a 3x3 Cauchy Green tensor. The eigenvalues of this matrix only take into account the relative deformation of an ellipsoid respect to their initial conditions, without distinction between horizontal or vertical movement. If the vertical deformations have the same weight than horizontal deformation, this could lead to mask the FTLE values.

Title: the subtitle highlights the influence of atmospheric rivers on large scale mixing variability, suggesting a causality (from ARs to mixing) which however is not clear to me in the results. In fact, by reading the manuscript, one has the feeling that the opposite may be also implied. The title should also take into account the fact that Lyapunov exponents are computed for the low troposphere only.

We agree with the referee that the title could lead to some confusion so we changed it to,

Climatology of Lyapunov exponents: The link between atmospheric rivers and large-scale mixing variability.

Please also note the supplement to this comment:
http://www.earth-syst-dynam-discuss.net/esd-2017-1/esd-2017-1-AC2-supplement.pdf